# A thorough analysis of the contribution of experimental, derived and sequence-based predicted protein-protein interactions for functional annotation of proteins

**Stavros Makrodimitris**[1,2]*, **Marcel Reinders**[1,3], **Roeland van Ham**[1,2]

**1** Delft Bioinformatics Lab, Delft University of Technology, Delft, the Netherlands, **2** Keygene N.V., Wageningen, the Netherlands, **3** Leiden Computational Biology Center, Leiden University Medical Center, Leiden, the Netherlands

* s.makrodimitris@tudelft.nl

## Abstract

Physical interaction between two proteins is strong evidence that the proteins are involved in the same biological process, making Protein-Protein Interaction (PPI) networks a valuable data resource for predicting the cellular functions of proteins. However, PPI networks are largely incomplete for non-model species. Here, we tested to what extent these incomplete networks are still useful for genome-wide function prediction. We used two network-based classifiers to predict Biological Process Gene Ontology terms from protein interaction data in four species: *Saccharomyces cerevisiae*, *Escherichia coli*, *Arabidopsis thaliana* and *Solanum lycopersicum* (tomato). The classifiers had reasonable performance in the well-studied yeast, but performed poorly in the other species. We showed that this poor performance can be considerably improved by adding edges predicted from various data sources, such as text mining, and that associations from the STRING database are more useful than interactions predicted by a neural network from sequence-based features.

## Introduction

One of the main challenges of the postgenomic era is how to extract functional information from the vast amount of sequence data that are available. As the number of known protein sequences grows at a very fast pace (currently >185 million in UniProtKB), experimentally determining the functions of all proteins has become practically infeasible. This creates the need for accurate Automatic Function Prediction (AFP) methods, which can predict a protein's function(s) using the knowledge that has been accumulated in the past. To this end, the Gene Ontology (GO) is a very valuable resource that provides a systematic representation of function in the form of three ontologies: Biological Process (BP), Molecular Function (MF) and Cell Component (CC) [1].

The Critical Assessment of Functional Annotation (CAFA) is a community-driven benchmark study that compares a large number of available AFP methods in an independent and

**Data Availability Statement:** All data used are from the public domain. Links to download and

instructions are provided at https://github.com/stamakro/revival-ppi.

**Funding:** This work was supported -in part- by Keygene N.V., an AgBiotech company in the Netherlands. The funder provided support in the form of salaries for authors SM and RvH, but did not have any additional role in the study design, data collection and analysis, decision to publish, or preparation of the manuscript. The specific roles of these authors are articulated in the 'author contributions' section. There was no additional funding (external or internal) received for this study.

**Competing interests:** Authors SM and RvH are affiliated with a commercial company. This does not alter our adherence to PLOS ONE policies on sharing data and materials.

systematic way [2–4]. One of the main conclusions that one can draw from the several editions of CAFA is that top-performing methods tend to use a combination of different data sources and not only the amino acid sequence. For example, MS-kNN, one of the best methods in CAFA2, combined sequence similarity with human gene co-expression and protein-protein interaction (PPI) data [5]. GOLabeler, which was the best in CAFA3, combined six different data sources with a powerful algorithm that predicts how suitable a GO term is for the input protein [6]. More recently, the authors of GOLabeler introduced an extension named NetGO which also uses PPI networks as an extra data source, reporting even better performance than GOLabeler on the CAFA3 dataset [7]. These observations show that PPI networks are informative data sources for AFP, which can be understood, since if two proteins physically interact, they are likely to be involved in the same biological process or pathway.

However, almost all PPI networks are incomplete. The best-characterized model species, *Saccharomyces cerevisiae* (baker's yeast), has one of the densest PPI networks, with 116,209 experimentally-derived, physical interactions in the BIOGRID database [8]. Given the fact that *S. cerevisiae* has about 6,000 protein-coding genes [9], this means that roughly 0.6% of all possible pairs of proteins are known to interact. The human interactome is also quite well characterized, with 424,074 experimental interactions in BIOGRID (about 0.2% of all possible interactions). Moreover, a recent study identified an additional 52,569 high-quality interactions of 8,275 human proteins [10]. On the other hand, in *Arabidopsis thaliana*, the most well-studied plant species, there are about 27,000 protein coding genes and 48,786 experimentally-derived physical interactions in BIOGRID, i.e. only 0.01% of the possible interactions are known. This is not likely due to protein interactions being less common in *A. thaliana*, but rather because it is not as well-studied as yeast.

The number of known edges is orders of magnitude smaller in other plant species, even in important crops. For example, in tomato (*Solanum lycopersicum*), there are only 107 interactions in BIOGRID as of June 2019 (<<0.01% of the total number of possible interactions). In rice (*Oryza sativa japonica*), there are 330 and in corn (*Zea mays*) 13. This phenomenon is not restricted to plants, but is also true for non-model animal species, such as economically important species like cow (*Bos taurus*, 529) and pig (*Sus scrofa*, 88 interactions).

Most methods that employ PPI networks in AFP predict functions by propagating the GO annotations through the network [5, 7]. The simplest of such methods transfers the annotations of a protein to its immediate neighbors. This is also known as Guilt-By-Association (*GBA*). Fig 1a illustrates the *GBA* method in an example network with 6 proteins: Proteins 1 and 2 are annotated with a GO term, while protein 6 is not. We are asked to predict whether proteins 3-5 should be annotated with that GO term. As seen in Fig 1a, for all three of these proteins we are at least 66.6% certain that they should be assigned that GO term. Fig 1b shows the same example network, assuming that some of its edges are missing. In this case, protein 5 has no known interacting partners, so it is impossible to determine its function. Similarly, protein 1 has a known function, but is disconnected from the rest of the network, so its function cannot be propagated to other proteins. This example shows that when interactions in a PPI network are missing, function prediction cannot benefit from PPI information (as most proteins will have few or no connections to other proteins).

A way to counter the lack of edges is to predict them using other data sources. The STRING database contains a large collection of protein associations predicted using different sources, such as gene co-expression and text mining [11]. Moreover, the recent rise in popularity of deep learning has caused an increase in methods that attempt to predict protein-protein interactions purely from protein sequence. One of the first examples was from Sun et al. [12], followed by DPPI [13], PIPR [14] and the work of Richoux et al. [15]. The advantage of predicting edges from sequence is that it is—at least in theory—not biased towards previous

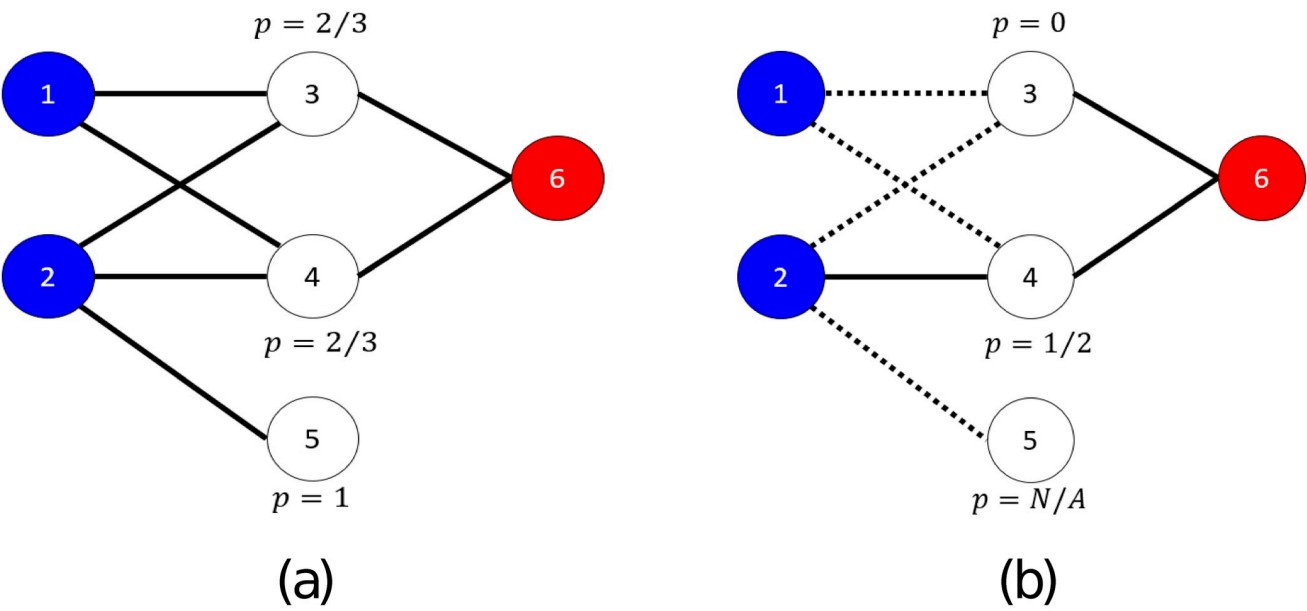

**Fig 1. Toy PPI network with 6 nodes.** Nodes annotated with a GO term are shown in blue and nodes not annotated in red. Unlabeled (test) nodes are shown in white. In (a) the entire network is known and the posterior probabilities for each unlabeled node can be calculated accurately. In (b) some of the edges are missing (signified by the dashed lines), making the calculation of posterior probabilities either erroneous or even impossible (e.g. node 5).

experiments. In contrast to, for example, predictions within the STRING database that still require other people to have previously studied a specific protein or its orthologues. Having an accurate sequence-based predictor of PPIs means that for all possible pairs of proteins we can obtain a score for how probable an interaction between each pair of proteins is. This would enable us to find possible interacting partners for proteins that have not been previously studied at all.

In this study, we are interested in quantifying the influence of missing edges in a PPI network on protein function prediction. Moreover, we are interested in how well (deep learning based) sequence-based PPI predictors can recuperate this missing information, and how that translates in improvements of the function prediction. We hypothesize that using such a model to predict interactions would be more effective than STRING in the downstream task of network-based protein function prediction.

## Materials and methods

### Protein-protein interaction networks

We compared PPI networks in *S. cerevisiae*, *Escherichia coli*, *A. thaliana* and *S. lycopersicum* using three types of PPIs: 1) Physical interactions that have been experimentally derived. 2) Predicted interactions based on non-experimental protein association data from the STRING database, and 3) Sequence-based predicted interactions based on the amino acid sequence of two proteins using PIPR.

**Physical interactions.** For the experimental interactions we used the BIOGRID (version 3.5.171) [8] and STRING databases [11]. We only used physical interactions and ignored the genetic interactions. Of note, the STRING database contains a collection of experimental protein-protein interactions from different databases, including BIOGRID (marked with the "experiments" data source code) and we found edges in BIOGRID that

were not present in STRING. From STRING, we only chose experimental protein-protein interactions with association scores larger than the median score over the non-zero scores for each species individually. The node degree distributions of these networks are shown in S1 Fig in S1 File.

**Predicted interactions.** Besides the experimental evidence, STRING contains protein associations from 12 data sources in total: "neighborhood", "neighborhood transferred", "co-occurrence", "database", "database transferred", "experiments transferred", "fusion", "homology", "co-expression", "co-expression transferred", "text mining" and "text mining transferred". We use these data as features predictive of two proteins interacting and/or being functionally associated to add edges to the experimental network. We refer to these edges as "predicted edges". S1 Table in S1 File shows the number of interactions per species and per data type. In each species, we ignored data sources that did not add any new edges. We also removed "database", as it includes protein associations that were identified by using the GO annotations of proteins and these edges would cause circular reasoning if used to predict GO terms, leading to a biased evaluation. This left us with 9 data sources from which we could infer PPIs in yeast, *E. coli* and *A. thaliana* and 8 in tomato (S1 Table in S1 File). The interaction scores have different distributions in different data sources. Therefore, instead of applying a fixed threshold, we selected the protein pairs with the 50% highest non-zero scores for each data source and species individually. Next to individually using the data sources as proxies for the protein-protein interactions, we also combined data sources. This was done by first integrating the STRING scores from different sources as described in [16] (see S1 File for more information) and then keeping the 50% top non-zero scores for every combination, as before. To combine a binary STRING network with the experimental one, we applied an element-wise logical OR to the corresponding adjacency matrices, so an interaction is added to the combined network if it is present in at least one of the original networks.

We also examined the possibility of using all STRING edges by creating weighted graphs whose edge weights correspond to the STRING interaction scores. We then added these weighted graphs to the binary experimental network.

**Sequence-based predicted interactions.** We used PIPR [14] to predict PPIs from protein sequence. It uses a Siamese twin architecture with both convolutional and recurrent units and three fully connected layers at the end. PIPR also makes use of predefined amino acid embeddings, obtained from both chemical properties of amino acids and their co-occurence in protein sequences. PIPR had an accuracy of about 97% in predicting yeast PPIs when trained on a large, balanced dataset from the DIP database. After having trained the model, we feed it all pairs of proteins. For each pair we get a score in the range [0, 1] denoting the probability that these two proteins interact. We add an edge to our predicted PPI network if the score for that edge is greater than or equal to 0.5.

## GO annotations

We obtained GO annotations from the GOA website [17] and only used the experimental annotations and curated annotations (evidence codes "EXP", "IDA", "IPI", "IMP", "IGI", "IEP", "IBA", "IBD", "IKR", "IRD" and "TAS"). We used the entire GO graph (not the smaller GO slim versions). Annotations were propagated towards the ontology root, so that when a protein is annotated with a term, it is also annotated with all its ancestors in the GO graph. We focused on the Biological Process Ontology (BPO), as it is the most difficult ontology to predict [3] and also is the most commonly used in further analyses such as gene set enrichment. Table 1 gives an overview of the different dataset sizes for the four species.

**Table 1. Number of proteins and known PPIs per species in BIOGRID. (version 3.5.171).**

|  | Yeast | E. coli | Arabidopsis | Tomato |
|---|---|---|---|---|
| approximate #protein-coding genes | 6,000 [9] | 4,400 [18] | 27,029 [19] | 34,727 [20] |
| #proteins with BPO annotations (N) | 4,997 | 2,869 | 10,648 | 651 |
| #BIOGRID edges between proteins with BPO annotations | 149,659 | 17,540 | 23,371 | 57 |
| #pairs of proteins with BPO annotations (N(N − 1)/2) | 12,482,506 | 4,114,146 | 56,684,628 | 211,575 |
| % annotated protein pairs interacting | 1.20 | 0.43 | 0.04 | 0.03 |
| % disconnected proteins | 0.4 | 23.1 | 43.4 | 96.9 |

## Function prediction methods

We represent the protein-protein interactions as a network with the proteins as nodes and the interactions as binary, undirected edges. Using this network, we can make predictions about the functions of unannotated proteins using the proteins with known function. To do so, we used a simple Guilt-By Assosciation (GBA) method and a more complicated one that uses node embeddings learned using *node2vec* [21]. We compared these methods to the *BLAST* and *naive* baselines, which are commonly used in the CAFA challenges [2, 3]. Each method computes the probability $P(p_i, t)$ that a GO term $t$ should annotate protein $p_i$. Below we provide details about how each method makes this computation. When $P(p_i, t)$ is undefined, e.g. because a protein has no neighbors in a PPI network or no significant *BLAST* hits, we set it to zero to indicate that this term cannot be assigned to this protein.

**Guilt-By-Association (GBA).** This method assigns a GO term to a protein with posterior probability equal to the fraction of the protein's interacting partners annotated with that term. More formally, let $A$ be the network's adjacency matrix, $V_{train}$ a set of training proteins and $V_{test}$ a set of test proteins. Moreover, let $T(p)$ be the set of GO terms assigned to $p \in V_{train}$. For a protein $p_i \in V_{test}$, we define its neighborhood $N(p_i)$ as all its interacting partners that are in the training set:

$$N(p_i) = \{p : p \in V_{train} \wedge A[p, p_i] = 1\} \tag{1}$$

For a GO term $t$, the probability it is assigned to test protein $p_i$ is given by Eq 2:

$$P(p_i, t) = \frac{\sum_{p \in N(p_i)} I(t \in T(p))}{|N(p_i)|} \tag{2}$$

Where $I(x) = 1$ *iff* $x$ is a true statement and $|S|$ denotes the number of elements in set $S$.

For weighted graphs, Eq 2 was adapted so that each neighbor transfers its annotations with a weight equal to the edge weight and we divide by the total sum of the weights instead of the number of neighbors.

**node2vec.** The *node2vec* algorithm learns a fixed-length embedding for every node, such that the similarity in the embedding space reflects the similarity of neighborhoods in the graph, as defined by random walks [21]. We used these embeddings as feature vectors on which we applied standard machine learning methods; specifically the $k$-Nearest Neighbors ($k$NN) and the ridge classifiers. For $k$NN, we look for the $k$ training proteins with the most similar feature vectors to a query protein $p_i$ and set $P(p_i, t)$ equal to the fraction of these $k$ proteins annotated with $t$. The ridge classifier models protein function prediction as a multi-output regression problem and learns a linear mapping from the feature space to the label space. We use $\mathbf{X} \in \mathbb{R}^{N \times d}$ to denote the *node2vec* feature matrix, where each row contains the feature vector of one protein, and $\mathbf{Y} \in \{-1, 1\}^{N \times L}$ to denote the label matrix, where each row represents the GO annotations of each protein and a value of 1 in the matrix denotes that the

corresponding protein is annotated with the corresponding GO term. The ridge classifier tries to find a linear mapping $W \in \mathbb{R}^{d \times L}$, such that $\mathbf{Y} \approx \mathbf{XW}$. We also add L2 regularization to the model with coefficient λ which leads to the optimal solution $\mathbf{W}^* = (\mathbf{X}^T \mathbf{X} + \lambda \mathbf{I})^{-1} \mathbf{X}^T \mathbf{Y}$. To bring the predictions ($\mathbf{XW}^*$) in the range [0, 1], we apply a sigmoid function $s(a) = (1 + e^{-a})^{-1}$ to each predicted value $a$. We did not post-process the predictions of the ridge method so it is possible that it makes predictions that are inconsistent with the GO hierarchy.

**Naive.** The *naive* method of CAFA [2] assigns a GO term to a protein with probability equal to the fraction of training proteins annotated with that term (Eq 3).

$$P(p_i, t) = \frac{|\{p : p \in V_{train} \wedge t \in T(p)\}|}{|V_{train}|} \tag{3}$$

This means that all test proteins get the same annotation using this method (making it a quite weak baseline).

**BLAST.** We ran *BLAST* with default settings and set $P(p_i, t)$ equal to the maximum sequence identity between $p_i$ and its hits annotated with $t$.

**Combining two classifiers.** Given the posterior probabilities of two classifiers $P_1(p_i, t)$ and $P_2(p_i, t)$ we combined them using Eq 4, which gives a high score for a protein-term pair if at least one of the two methods gives a high score.

$$P_{combo}(p_i, t) = 1 - (1 - P_1(p_i, t)) \cdot (1 - P_2(p_i, t)) \tag{4}$$

## Experimental set-up

**Evaluation metrics.** To compare function prediction across the differently constructed protein-protein interaction networks, we applied a 5-fold cross-validation. As evaluation metrics we used the protein-centric $F_{max}$ and $S_{min}$ that are extensively used in the CAFA challenges. Definitions for these metrics are provided S1 File. We also measured the coverage of each algorithm, defined as the fraction of test proteins for which at least one term has a non-zero posterior probability.

As the GO term distributions and frequencies are different in each species, directly comparing the performances across species is not trivial. To counter the effect of GO term frequencies, we use the concept of Prediction Advantage (PA) [22], which is defined as the improvement on the classification loss of a classifier $c$ ($L_c$) with respect to the *naive* classifier ($L_{naive}$). The PA, which is defined in Eq 5, can be calculated for any classification loss, so here we used $L = 1 - F_{max}$.

$$PA(c, L) = 1 - \frac{L_c}{L_{naive}} \tag{5}$$

In each fold, we discarded the GO terms that had no positive examples in either the training or the test set.

**Experimental PPI (*EXP*).** We started from the experimental PPI network of a given species. This network includes as nodes all proteins that have at least 1 functional annotation, even if they have no interacting partners. Proteins without functional annotations were removed, even if they had known interactions.

*node2vec* is an unsupervised feature extraction step that only depends on the network and not the functional annotations. We additionally tested whether also including the unannotated proteins as nodes in the network would possibly lead to better features in the first step of the *node2vec* procedure, as it leads to a better neighborhood estimation. To this end, we ran

*node2vec* on the entire *EXP* network (including unannotated proteins) and then used the extracted (*unsupervised*) features of the annotated proteins only in the *supervised* phase. We repeated this experiment for all four species and compared the performance with that of the original *node2vec* which learned the (unsupervised) features on a network of only annotated proteins.

**Combined experimental and predicted PPI (*EXP+STRING*).** We added predicted edges to the experimental network from the different data sources in STRING. We evaluated all possible combinations of the 9 STRING data sources (8 for tomato): First, we added each data source individually. Then, we tested all combinations of 2 data sources (36 possibilities), all combinations of 3 (84 possibilities) and so on, until we have included all 9 data sources. So, in total, we tested $\sum_{i=1}^{9} \binom{9}{i} = 511$ combinations of data sources (255 for tomato) along with the experimental network.

**Sequence-based predicted PPI (*EXP+SEQ*).** We used edges predicted by PIPR for predicting function. We tested the performance of a network with the experimental edges combined with the PIPR predictions.

**Optimization of *node2vec* classification.** *node2vec* has hyperparameters that can have a large influence on the learned features. We tuned these hyperparameters on the experimental PPI network of each species, by splitting the training set of each cross-validation fold into a new training (80% of initial training set) and a validation set (20% of intial training set). For each hyperparameter combination, we generated node features which we fed to the *k*NN and ridge classifiers for different values of their parameters (*k* and λ respectively). Finally, for each cross-validation fold, we identified the combination of hyperparameters, classifier and classifier parameter that maximized the $F_{max}$, trained it on the whole training set and used the trained model to make predictions on the test set. Details about the hyperparameters that were tuned and the values considered are provided in S1 File.

When running *node2vec* on all proteins with known interactions (and not only the ones with functional annotations), we again used 5-fold cross-validation as before. The training, validation and test splits in each fold were kept identical. We also repeated the hyperparameter optimization step, as changes in the network topology might call for different hyperparameter values.

## Results

### Only the yeast experimental PPI network has acceptable function prediction performance

Fig 2a–2d compare the $F_{max}$ achieved by the *GBA* method on the *EXP* network to the baseline performances in four species using 5-fold cross-validation. In yeast, this simple approach significantly outperforms both *naive* (p-value $< 10^{-5}$, paired t-test, FDR-corrected) and *BLAST* (p-value $= 0.5 \cdot 10^{-3}$, paired t-test, FDR-corrected). In *E. coli*, *A. thaliana* and tomato, the picture is quite the opposite, with even the *naive* method largely outperforming *GBA* (p-values $= 0.026, 0.3 \cdot 10^{-5}, 0.2 \cdot 10^{-3}$ respectively, paired t-test, FDR-corrected). In tomato, the network is so sparse and disconnected that the maximum F1 score is achieved by assigning all GO terms to all proteins. The Prediction Advantage (PA, see Methods) between *GBA* and *naive* classifier follows a linear trend with respect to the fraction of existing edges. The calculation was based on only four points, but it still lies under the statistical significance threshold of 0.05 (Fig 2e, Pearson's $\rho = 0.98$, p-value $= 0.016$).

To better characterize the effect of missing edges, we simulated the phenomenon in yeast by removing edges either uniformly at random or by an approach that makes nodes with the lowest degree more likely to lose their edges first (S1 File). We found that the $F_{max}$ is relatively

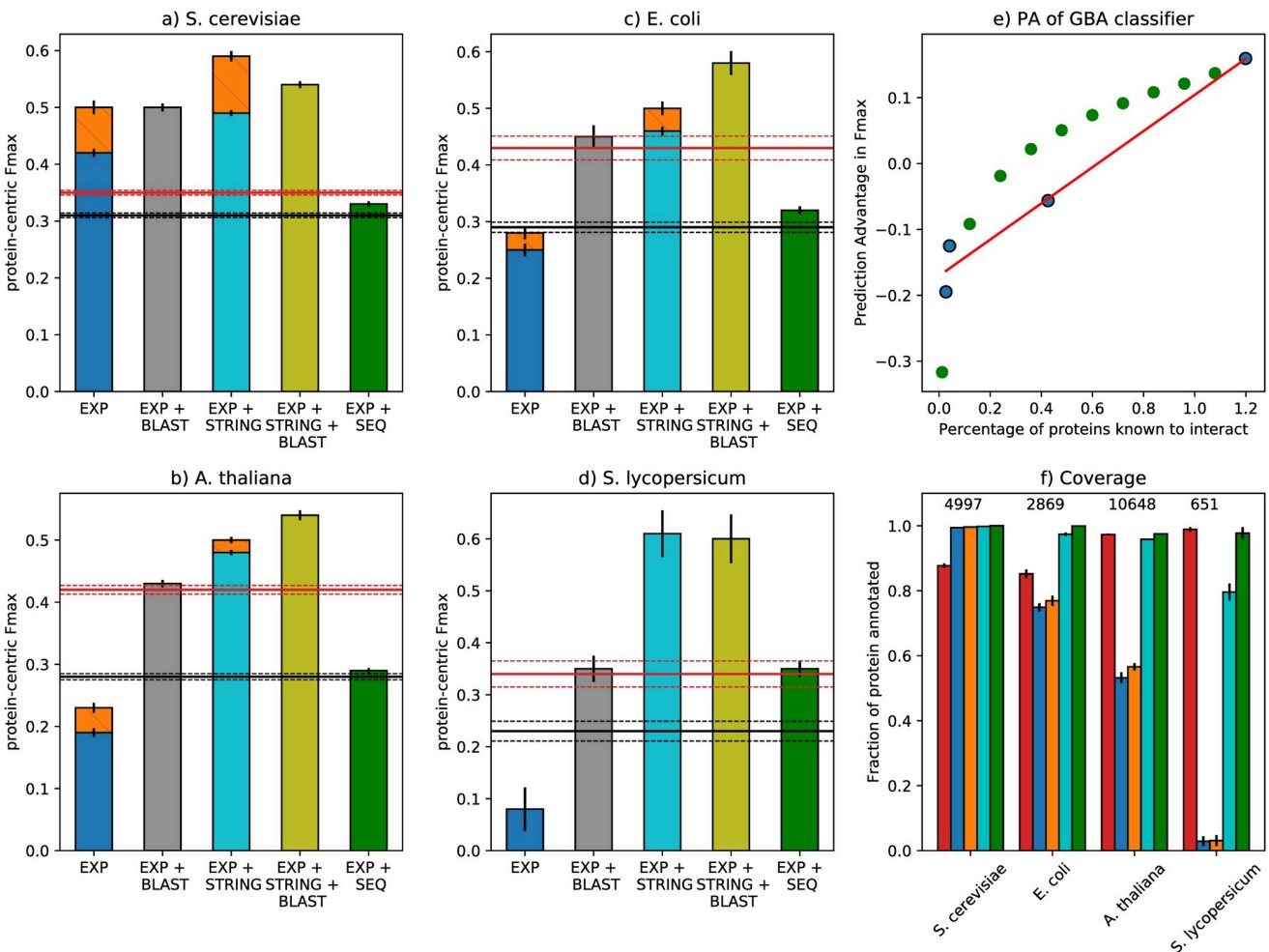

**Fig 2. Function prediction performance of PPI networks in four species.** (a-d): On the *x*-axis, are the different PPI networks. The height of the bars denotes the $F_{max}$ in each species. The *naive* and *BLAST* baselines are shown as a red and a black horizontal line respectively, with dashed lines showing the corresponding standard deviations. *EXP*, *GBA* is shown in blue, *EXP+STRING*, *GBA* in cyan and *EXP+SEQ* in green. The improvement of *node2vec* on *EXP* and *EXP+STRING* is shown as an orange bar. Absence of an orange bar denotes that the two algorithms performed equally. The combinations of *EXP*, *GBA* and *EXP+STRING*, *GBA* with BLAST are shown in gray and yellow respectively. The error bars denote the standard deviation over the 5 cross-validation folds. e) Prediction Advantage (PA) of $F_{max}$ as a function of the fraction of known interactions. Each species is shown as a blue dot and red line shows the least squares linear fit. PA values calculated by downsampling the original yeast network at different levels of missing edges are shown as green dots. f) The fraction of annotated proteins for which each method can make predictions (y-axis) for each species (x-axis). On top, the number of total proteins is shown. Different methods are shown in the same colors as in a-d. Note that the naive method has a coverage of 100% by design.

robust to uniform edge removal up to 40-50%, but $S_{min}$ deteriorates more quickly (S2 Fig in S1 File), meaning that predicting more specific terms suffers even under this simplified missing edges scenario. The coverage also drops very slowly (at least initially), which implies that most edges are removed from "dense" parts of the network so that the remaining edges can partly make up for this loss. In the degree-based sampling strategy, which is more realistic, we observed a much steeper drop for all three metrics. In this case, poorly-studied proteins lose their connections very quickly making it impossible to make predictions for them, as indicated by the steep decline in coverage. As a result, the average performance also reduces very fast. The PA values calculated from the degree-based downsampling did not confirm the linear relationship between PA and fraction of known edges (green dots in Fig 2e).

**Combining PPI networks with homology.** In many function prediction pipelines, PPI networks are combined with other data sources and used in ensemble algorithms. Experiments with a simple method that fuses the posterior probabilities of *BLAST* with those of the PPI classifier (Eq 4) showed minimal performance gains (2-6%) with respect to stand-alone *BLAST*, for all species except for *S. cerevisiae* (43%, Fig 2). The difference with respect to *BLAST* was found statistically significant using the paired t-test. However, after correcting for multiple testing using the False Discovery Rate method, the p-values for *E. coli*, *A. thaliana* and tomato lie just below the 5% significance threshold (0.0468, 0.0468 and 0.0486 respectively), whereas for yeast the corrected p-value is $1.5 \cdot 10^{-5}$. These results confirm that using experimental PPI networks with many missing edges is not helpful for function prediction.

*node2vec* **results.** The *GBA* method is very simple and therefore unlikely to be able to capture all the functional signal present in complicated biological networks. We therefore tested whether a more complicated classifier based on *node2vec* could outperform it. In the same cross-validation loop, we used a validation set to tune the hyperparameters of *node2vec* and used the same unseen test set as before to evaluate the model. The optimal hyperparameter values varied per cross-validation fold and per species. The *1*NN classifier was the optimal choice in yeast and tomato, while the ridge with moderate regularization in *E. coli* and *A. thaliana*. More importantly, *node2vec* performed better than *GBA* on the *EXP* network in all species except for tomato, where assigning all terms to all proteins still maximizes the $F_{max}$ (Fig 2a–2d, S2 Table in S1 File). Evaluation based on $S_{min}$ gave similar results (S3 Table in S1 File).

We also tested whether including proteins with known interactions but no functional annotations during the feature learning step could improve the performance of *node2vec*. We used the t-test to compare the $F_{max}$, $S_{min}$ and coverage of these networks to the ones that consist of only annotated proteins. We found that doing so lead to a small but significant increase in coverage in *E. coli* and *A. thaliana* (paired t-test, corrected for the FDR), but there was no significant difference in $F_{max}$ or $S_{min}$ in any of the four species (FDR > 0.05, S4 Table in S1 File). This means that although we can make predictions for more proteins the predictions become less accurate when including these edges. Therefore, for the rest of our experiments we only refer to *node2vec* trained on the proteins that have GO annotations.

**Performance per protein.** Comparing the performance for each individual protein, we observed a large non-linear dependency between the performance and the number of annotated neighbors. This dependency was consistently smaller for *node2vec* (a Spearman correlation of 0.30, 0.60 and 0.81 for yeast, *E. coli* and *A. thaliana* respectively) than for *GBA* (0.41, 0.65 and 0.85 for yeast, *E. coli* and *A. thaliana* respectively). We also found that *node2vec* consistently outperforms *GBA* regardless of the number of annotated (training) neighbors in *E. coli* and *A. thaliana* (Wilcoxon rank sum test, FDR < 0.05, Fig 3 and S4 Table in S1 File). In *S. cerevisiae*, *node2vec* is significantly better than *GBA* for 6 out of 9 bins and significantly worse in 1 bin, while for two bins there were no significant differences (Wilcoxon rank sum test, FDR < 0.05, Fig 3 and S5 Table in S1 File). Finally, *node2vec* can make predictions for proteins that do not have any training neighbors as long as they are not completely disconnected, as its feature vectors are learned in an unsupervised way using the entire network. This means that, for not too sparse networks, *node2vec* is the preferred option compared to *GBA*.

## Adding predicted edges is more useful than using a complex classifier

We then tested to what extent predicted interactions from STRING can improve upon the protein function prediction performance of the *EXP* networks. As we can see in Fig 2a–2d, the *GBA* classifier performed considerably better on the *EXP+STRING* network than on *EXP* for all species. It also significantly outperformed the *naive* and *BLAST* baselines. As shown in

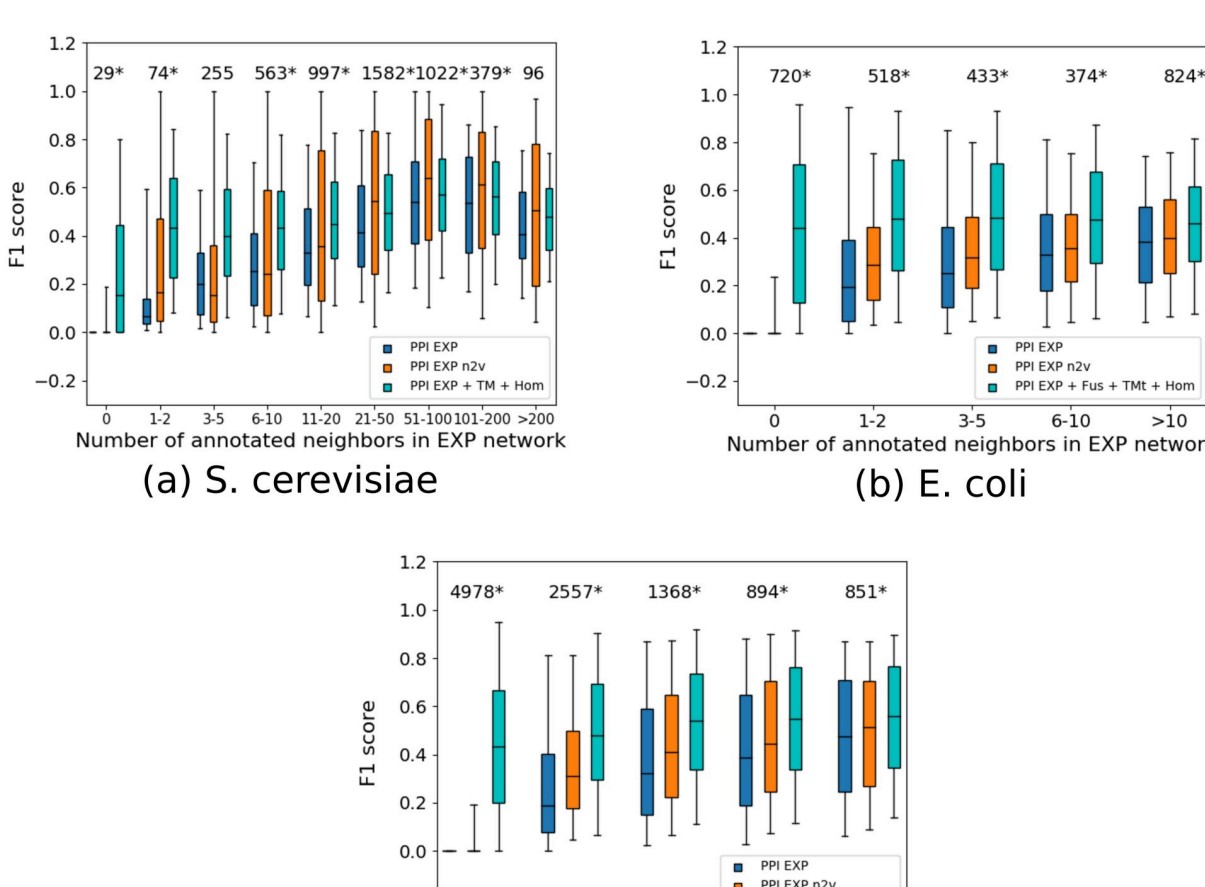

**Fig 3. Performance per protein.** $F_{max}$ achieved per protein ($y$-axis) as a function of the number of training neighbors in the *EXP* network ($x$-axis) for *EXP*, *GBA* (blue), *EXP*, *node2vec* (orange) and *EXP+STRING*, *GBA* (cyan). The median of each group is denoted by a horizontal line and the 5th and 95th percentiles by the whiskers. The number of proteins in each group is shown at the top of each group and an asterisk (*) next to the number signifies that the difference between *EXP*, *GBA* and *EXP*, *node2vec* is statistically significant at a False Discovery Rate of 5%. For the *EXP+STRING* network, we show the performance of the combination of data sources that had the best performance in each species.

Fig 3, the STRING edges offer a performance boost for both nodes that have and nodes that do not have annotated neighbors in the experimental network for all species. However, for hub yeast proteins with more than 20 experimental edges, applying *node2vec* on the *EXP* network was more effective than adding predicted edges (Fig 3a). The fraction of proteins that can be annotated by the *STRING* networks approaches 100% for *E. coli* and *A. thaliana* and 80% for tomato (Fig 2f).

Using a weighted STRING network with all available interactions instead of a binary one lead to small performance improvements, but mainly for the combinations that performed less well (S3–S6 Figs in S1 File). The effect sizes were rather small for the top-performing combinations (S6 Table in S1 File). This shows that STRING edges possibly contain useful functional signal even at confidence levels lower than those we considered here.

**Combining STRING edges with homology.** Moreover, combining the predictions of the *GBA* classifier on this network with *BLAST* predictions (see Methods) leads to significant improvement (28-76%) over *BLAST* for all species (Fig 2). The combined model gave

significant improvements (10-26%) over its PPI component in yeast, *E. coli* and *A. thaliana* and performed equally well in tomato (Fig 2). $S_{min}$ results show similar trends, with the exception that in yeast, the optimal $S_{min}$ is achieved by *GBA* on the *EXP+STRING* network and not by the combination with *BLAST* (S3 Table in S1 File). These show that adding predicted edges is very beneficial for all tested PPI networks.

*node2vec* **on STRING edges.** Similar to the *EXP* network, we compared the *GBA* classifier to the one based on *node2vec* on *EXP+STRING*. We again observed that the more complex classifier achieved higher $F_{max}$ in yeast, *E. coli* and *A. thaliana* (Fig 2a–2d), but in terms of $S_{min}$ only yeast showed an improvement (S3 Table in S1 File). In addition, Fig 2b–2d show that in not so well-studied species, using a more complicated classifier on the *EXP* network performs considerably worse than a simple classifier on a more complete network with predicted edges.

**Effect of individual STRING data sources.** We also examined which *STRING* data sources were responsible for the observed increase in performance. As shown in Fig 4 and S7–S9 Figs in S1 File, the vast majority of data sources when individually added to the *EXP* network lead to better function prediction in terms of both $F_{max}$ and $S_{min}$, with the exception of "experiments transferred" in yeast. Fig 4 and S7–S9 Figs in S1 File also show that "text mining" (in *S. cerevisiae* and *A. thaliana*), "text mining transferred" (in *E. coli* and *S. lycopersicum*) and "homology" (in all four) were by far the most useful sources. A more in-depth analysis of the results showed that these three data sources alone are actually enough to obtain the maximum performance of the *GBA* method on the *EXP+STRING* network (S7–S14 Tables in S1 File) and that removing all of them leads a to significant performance drop (S15 and S16 Tables in S1 File). Moreover, including all nine data sources (eight for tomato) lead to worse $F_{max}$ and $S_{min}$ in all species (Fig 4 and S7–S9 Figs in S1 File).

## Edges predicted from protein sequences by a neural network are less useful than STRING edges

The *PIPR* model for predicting protein-protein interactions from sequence was reported to have 97% cross-validation accuracy on a balanced dataset with about 11,200 data points from *S. cerevisiae* proteins from the DIP database, a result that we also replicated. This model, however, was not able to generalize to predict BIOGRID edges in yeast, as it achieved an accuracy of 0.59 on a balanced dataset. We also measured the model's recall, i.e. its ability to identify true interacting pairs, and it was comparable to random guessing (0.51).

We therefore set out to train PIPR for predicting BIOGRID edges, keeping the architecture and the training procedure the same. As positive training examples, we used all yeast protein pairs reported to be physically interacting in BIOGRID and as negative examples, an equal-sized set of randomly selected protein pairs that are not reported as interacting. This proved to be a more challenging task for PIPR, as the best validation accuracy achieved was 0.77 (S17 Table in S1 File).

The sequence-based predicted PPI network combined with the experimental one (*EXP+SEQ*) hampers the AFP performance in yeast as compared to *EXP* (Fig 2a). This is probably due to the addition of many false positive edges, as it predicts that more than 41% of all possible protein pairs are interacting, which is about 10 times more than expected [16]. In contrast, in *E. coli*, *A. thaliana* and tomato the *EXP+SEQ* PPI network seems to be more useful, providing significant improvements over *EXP* (Fig 2b–2d). However, these improvements are not enough to surpass even the *BLAST* baseline in *E. coli* and *A. thaliana*. Contrary to our expectation, the *EXP+STRING* network performed significantly better than *EXP+SEQ* for all species (Fig 2a–2d). This was true even when we removed edges from text mining from the *STRING* networks.

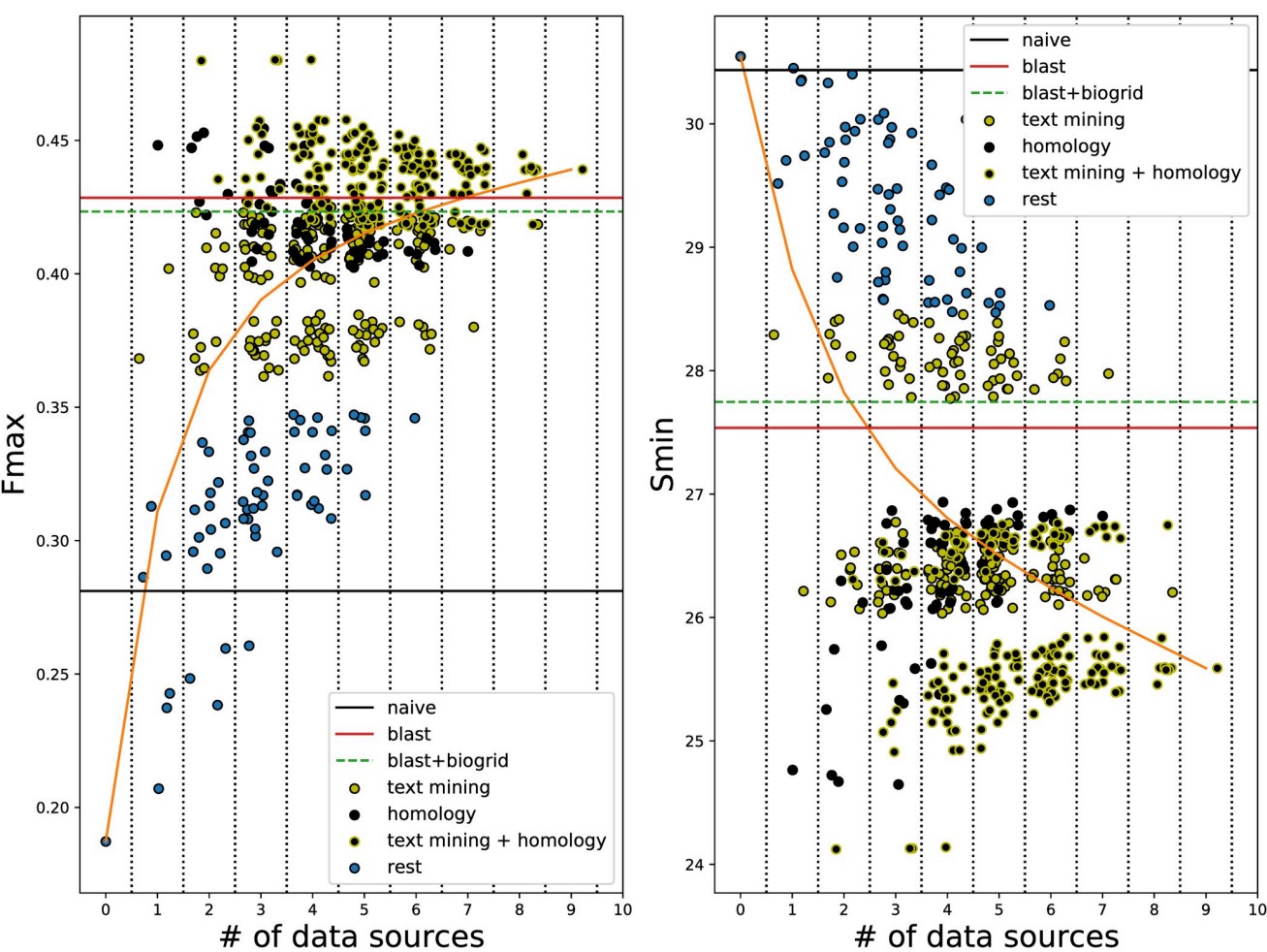

**Fig 4. Performance of STRING edges in *A. thaliana*.** $F_{max}$ (left) and $S_{min}$ (right) (*y*-axis) as a function of the number of *STRING* data sources included (*x*-axis). Each dot corresponds to one combination of data sources added to the experimental network. Combinations that include "text mining" and/or "text mining transferred" are shown in yellow, combinations that include "homology" in black and combinations that include both in black with yellow border. The rest of the combinations are shown in blue. To ease visibility, we added a random number in the range [-0.5, 0.5] to each combination of the same number of sources. Zero data sources corresponds to the *EXP* network and the orange line shows the average performance for a specific number of data sources. Horizontal lines denote the performance of the *naive* (black), *BLAST* (red) and the combination of *BLAST* with the *EXP* PPI network (dashed green).

In tomato, the *EXP+STRING* network cannot make predictions for roughly one fifth of the proteins (Fig 2f). Adding the *SEQ* edges only for these proteins improved the overall $F_{max}$ from 0.61 to 0.67. This shows that *SEQ* edges are useful, but they are surpassed by the higher quality of *STRING* edges.

Finally, we trained *PIPR* on *A. thaliana* edges from BIOGRID and obtained new networks in *A. thaliana* and *S. lycopersicum*. Although this network worked slightly better in tomato than the one trained in yeast data, it was still worse than *BLAST* and *EXP+STRING* (S18 Table in S1 File).

## Discussion

The aim of this work was to investigate ways of addressing the problem of missing edges in experimental protein-protein interaction networks for the downstream task of genome-wide function prediction. Our main hypothesis was that a deep learning model that can identify

interacting proteins from sequence with very high accuracy would be a good solution to this issue.

We demonstrated how the sparsity of experimental PPI networks leads to poor function prediction performance, using the simple *GBA* classifier. We did not compare this classifier to any state-of-the-art methods, such as GOLabeler [6] or INGA [23], but rather to the *naive* and *BLAST* baselines from the CAFA challenges. The *naive* classifier, as its name suggests, does not use any information to relate specific proteins to GO terms, rather it only uses the frequency of each GO term in the training set. In the machine learning literature, this classifier is also called the "Bayesian Marginal Predictor" [22] and is the optimal classifier when the distributions of the classes ($P(y)$) are known, but information about the relationship between the data and the classes ($p(x|y)$) is missing. This means that any classifier that uses any kind of (informative) data is expected to outperform the *naive* one.

However, we clearly demonstrated the failure of the *GBA* classifier in predicting BPO terms in *E. coli*, *A. thaliana* and tomato, as it performed considerably worse than the *naive* method. This was not the case in yeast, where the *GBA* classifier outperformed both baselines. When examining the performance for individual proteins, we found a high correlation between the number of known interacting partners and the prediction accuracy.

The *GBA* method has proven to be very useful in function prediction [5], but it is a very simple approach and therefore heavily relies on the correctness of the given network. We thus expected that using a more complicated approach that captures broader network patterns might (partly) overcome the sparsity. Several such node classification methods exist [24]. Recently, Graph Convolutional Networks (GCNs) have been shown to be effective in such tasks [25]. We chose to use *node2vec* to generate node features, as it has been successfully applied to protein-protein interaction networks [21] and used these features to train standard classifiers for function prediction. Although, we observed a clear improvement in *A. thaliana* and *E. coli* with respect to *GBA*, the performance remained below that of the baselines, meaning that these models can only partly compensate for missing edges. To make matters worse, we did not observe any improvement in the even sparser tomato network. This difference can be explained by the fact that when tuning the *node2vec* hyperparameters we rely on the performance on a validation set, which in tomato is very small and only includes "easy" proteins, leading to an apparent high performance for a large number of hyperparameter combinations. This makes it hard to select the optimal hyperparameters for *node2vec* in tomato, but it is still possible that an improvement could be observed if the correct parameters were known. Using the optimal hyperparameters from another species with a more complete network, e.g. A. thaliana, might be an alternative. However, since the topologies of the two networks are vastly different, the optimal hyperparameters for one species are not necessarily good for the other. Taken together, these observations validated our hypothesis that a sparse PPI network is detrimental to genome-wide AFP.

It is worth noting that *node2vec* can make predictions for nodes that have no annotated neighbors, as opposed to *GBA*, which helps increase the coverage. Nevertheless, including unannotated proteins with known interactions during the *node2vec* feature learning step did not lead to better function prediction performance. This hints that -apart from the lack of known interactions- the lack of GO annotations for training proteins also has a considerable negative effect on the accuracy of function prediction algorithms.

Many methods have been proposed that try to complete a network by predicting edges. Reviews of such methods can be found in [26] for social and in [27] for biomedical networks. More specifically, the computational prediction of protein-protein interactions has been an active research area for many years [28, 29]. Our work is the first to evaluate the contribution of predicted edges in protein function prediction in a species-specific way. We used the

STRING database as a proxy for predicting interaction using omics data such as genome features, homology, co-expression and text mining. In sparse experimental PPI networks, the *STRING*-derived edges contribute a great deal, increasing the performance of the *GBA* classifier 1.8-fold in *E. coli*, more than 2.5-fold in *A. thaliana* and about 30-fold in tomato. They also outperformed the *node2vec* method on the *EXP* network. This is because these extra edges connect proteins that were previously disconnected from the rest of the graph, but also because they can discover new functions for already connected proteins, leading to a performance boost regardless of the number of neighbors. Using these edges was enough to significantly outperform the *naive* and *BLAST* baselines. In the case of yeast, which has a more "complete" network, the *STRING*-derived edges also improved the prediction performance, but to a lesser extent. In fact, in yeast, *node2vec* on the *EXP* network and *GBA* on the *EXP+STRING* network performed similarly on average, with *node2vec* being more useful for hub proteins that have a complicated neighborhood. As expected, combining a better network (*EXP+STRING*) with a better classifier (*node2vec*) lead to even better performance, though this was not observed in the small tomato dataset.

To combine the different *STRING* data sources, we used the simple algorithm described in [16]. This algorithm (also described in S1 File) assumes independence between the data sources and applies a Bayesian framework to join them into a final score for each protein-protein association. Some more advanced methods have been proposed to perform this integration, such as *Mashup* [30] and *deepNF* [31]. Both of these approaches, which are conceptually similar to each other and to *node2vec*, perform a number of random walks separately for each network derived by each data source to estimate the neighborhood similarity of each node to all other nodes. Then, they learn a feature vector for every node (protein) in order to approximate this similarity as closely as possible. The main difference between the two methods is that *Mashup* learns these vectors using matrix factorization [30], while *deepNF* using an autoencoder neural network [31]. Both of these methods outperformed the simple integration strategy in yeast and human PPI networks [31], which means that the performance of the *EXP+STRING* network could be enhanced by using one of these two methods. On the other hand, these methods—and especially *deepNF* that has many parameters to be learned—are not guaranteed to work well in a small dataset such as the tomato one. Furthermore, as *STRING* networks have weighted edges, instead of using thresholds to make them binary, it might be more helpful to employ algorithms that classify nodes directly on weighted graphs, such as those described in [32] and [33]. Our small-scale experiments in that direction gave mixed results, so more research is needed on this issue.

Notably, text mining of scientific literature and homology were the most informative STRING data sources for all species. Although removing the text mining edges did lead to a decrease in the maximum performance of *EXP+STRING* networks, we showed that it did not change the main conclusions of this study. Moreover, we found that edges from "text mining transferred", i.e. associations that have been discovered through text mining in other species and then transferred based on sequence homology, are very useful in *E. coli* and tomato. Given that we did not consider GO annotations inferred automatically due to sequence similarity, it is likely that text mining indeed captures true functional information that is conserved across species. This perhaps means that text mining is an underrated data source for functional annotation. We hypothesize that since scientific knowledge is mainly disseminated by publishing articles, text mining on these articles compiles all of this information into one resource. This would explain why otherwise very informative resources such as gene co-expression or operons (in bacteria) are individually useful when added to the *EXP* network, but are rendered redundant in the presence of text mining edges. Although homology is the most commonly used data source for function prediction, from the descriptions of the methods submitted to

the CAFA challenges, we know that only a small minority of them make use of text mining [4]. Two of these methods are described in [34, 35]. A more recent study showed that integrating homology-based predictors with neural-network-based text models leads to a significant performance boost [36], so we expect the role of text mining in function prediction research to be expanded in the future.

We also applied a sequence-based neural network model (PIPR) for PPI edge prediction. Firstly, we noticed that although PIPR was very accurate in predicting edges in one yeast dataset, it did not immediately generalize to another dataset from the same species, performing very close to random guessing. Richoux et al. have reported that overfitting and information leaks from the validation set are common when training protein-protein interaction predictors [15]. Although a certain protein pair from the test set cannot be present in the training set too, the two individual proteins can be in the training set in other pairs. This can have an effect for hub proteins with many interacting partners, as in an extreme case the network could learn to always predict this protein to interact with any other protein [15]. The result of these findings as well as ours is that caution is required when using these deep models, despite their high accuracy in one dataset.

Nevertheless, PPIs predicted from the PIPR model can be useful for the downstream task of network-based function prediction, as they outperformed the *naive* baseline. However, our hypothesis that such a model could accurately produce the entire or a big part of the interactome of a species leading to very accurate predicted annotations was not validated, as *STRING* edges proved more useful. Our experiments in tomato showed that for proteins that were disconnected in the *EXP+STRING* network, adding *SEQ* edges gave a significant performance increase, while this was not the case for combining the *EXP+STRING* network with *BLAST*. This implies that *SEQ* can be a useful resource for species with very few protein associations known in *STRING*.

Another limitation of our study is that except for the variable degree of unknown PPIs among the tested species, there is also a large variability in the amount of missing experimental annotations, with yeast being the most well-characterized species and tomato by far the least. This means that it is much more likely that a correctly predicted protein-GO term pair is flagged as a false positive in tomato than in yeast, simply because that annotation has not been discovered yet. Moreover, the GO terms have different frequencies in the four species, meaning that is virtually impossible to compare performances across species. For example, yeast contains a lot more specific annotations than e.g. tomato. This is also demonstrated by the large differences in $S_{min}$ of the *naive* method, which means that the total information content of the terms present in each species is vastly different. Calculating the Prediction Advantage with respect to the *naive* method [22] can correct for differences in term frequencies, but the different degree of missing annotations is harder to correct for while only using experimental annotations. This is not a big issue in our analyses because we did not focus on the exact performance values, but rather on how the performances of different networks (i.e. networks with different edge types) compare to each other within a species. Also, we have shown that the same conclusions can be drawn when evaluation is done using the semantic distance [37], which punishes shallow predictions.

Although $F_{max}$ and $S_{min}$ are the most widely-used evaluation metrics for function prediction, a recent study has raised concerns about them [38]. The concerns, which were based on artificially generated predicted annotations, mainly have to do with these metrics being overly lenient to false positive predictions. This might not be a big problem, as due to missing annotations most proteins are likely to be under-annotated. The same study showed that both metrics correlate highly with the signal to noise ratio of the predictions [38]. Based on that we argue that our conclusions do not rely on the choice of evaluation measures, but we believe that

proper evaluation of function prediction algorithms is a pressing issue that requires further research.

## Conclusion

Our work highlights the difficulty of applying PPI networks in AFP for less well-studied species. We show that predicted PPIs can partially compensate for the sparsity of the networks, with STRING-predicted edges to be the most useful, especially text mining and homology, and sequence-based deep learned predictions mostly to be useful when nodes are still not connected when combining experimental and STRING based PPI edges.

## Supporting information

**S1 File.**
(PDF)

## Author Contributions

**Conceptualization:** Stavros Makrodimitris.

**Formal analysis:** Stavros Makrodimitris.

**Funding acquisition:** Roeland van Ham.

**Methodology:** Stavros Makrodimitris, Marcel Reinders, Roeland van Ham.

**Software:** Stavros Makrodimitris.

**Supervision:** Marcel Reinders, Roeland van Ham.

**Visualization:** Stavros Makrodimitris.

**Writing – original draft:** Stavros Makrodimitris.

**Writing – review & editing:** Marcel Reinders, Roeland van Ham.

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
