## [Decision Letter · Decision Letter 0]

10 Aug 2020

PONE-D-20-21508

A thorough analysis of the contribution of experimental, derived and sequence-based predicted protein-protein interactions for functional annotation of proteins

PLOS ONE

Dear Dr. Makrodimitris,

Thank you for submitting your manuscript to PLOS ONE. After careful consideration, we feel that it has merit but does not fully meet PLOS ONE’s publication criteria as it currently stands. Therefore, we invite you to submit a revised version of the manuscript that addresses the points raised during the review process.

Specifically, the computational experiments to address the extend to which Protein Protein Interaction Networks are useful in predicting Gene Ontology terms in different species are well thought and designed. Moreover, the work is well organized and presented in an intelligible manner. Both expert reviewers provide useful suggestions (see below for their detailed reports) which can strengthen the conclusions derived from this work and/or improve presentation.

I would also like to stress that you should make sure that all figures and tables are appropriately cited in the text (main text and/or supplement). For example, Tables S12 and S13 in Supplementary Text S1 are not cited either in the main manuscript or in Text S1.

We look forward to receiving your revised manuscript.

Kind regards,

Vasilis J Promponas

Academic Editor

PLOS ONE

Journal Requirements:

Reviewers' comments:

Reviewer's Responses to Questions

**Comments to the Author**

1. Is the manuscript technically sound, and do the data support the conclusions?

Reviewer #1: Partly

Reviewer #2: Yes

2. Has the statistical analysis been performed appropriately and rigorously? 

Reviewer #1: Yes

Reviewer #2: Yes

3. Have the authors made all data underlying the findings in their manuscript fully available?

Reviewer #1: Yes

Reviewer #2: Yes

4. Is the manuscript presented in an intelligible fashion and written in standard English?

Reviewer #1: Yes

Reviewer #2: Yes

5. Review Comments to the Author

Reviewer #1: The paper aims to perform an analysis of how PPI networks can help in the functional annotation of proteins. The authors also want to study the impact of the quality of a PPI (how well known they are).

To do so, they first compare naive algorithms to annotate function with guilt-by-association (GBA) and node2vec (n2v) algorithms using PPI. Then they compare GBA on different PPI: they combine the PPI with different STRING networks. Finally they compare those algorithms with a deep-learning method based on sequences. They do those comparisons on four organisms: yeast, with a well-known PPI, and A. thaliana, E. coli and tomato which do not have a well-known PPI.

The paper is well written and nice to read.

I have one major concern:

1) The way the "EXP" PPI is made: you chose to remove the nodes that were connected but without functional annotations. You say it yourself in the Discussion: this might have negatively influenced the performance. But you don't explain your choice of getting rid of those nodes. You could either let the nodes in the "EXP" PPI, or make another PPI with those nodes to see the difference.

You use algorithms that use neighborhood, then if you modify the neighborhood you can't expect them to perform well.

Also you consider STRING networks as "predicted interactions", but all of them are not:

- "neighborhood" just states if the genes occur repeatedly in close neighborhood in genomes (mostly prokaryotic). So this one is not a prediction, and might be useful only for E.coli.

- "co-occurence" shows the occurence of two genes across species, here again mostly prokaryotic species, and no prediction.

- "homology" is a score that is not used "as is" in STRING scoring schemes, it is likely a BLAST used in other channels.

- "text-mining" can not predict anything because it just extract information from published articles.

You do prediction by using this data as input to one of the algorithm you use, but STRING networks are not predicted.

I have several other comments:

2) You have a typo in the abstract: "Here, we tested to what extened" -> "to what extent"

3) You cite STRING v10, but on your github you say you used STRING v11.

4) You chose to select only the 50% highest non-zero scores, have you tried other percent? Why 50%?

5) In Table S2, you might have swap the "EXP, GBA" and "EXP, node2vec" tomato values, because on the text and on Figure 2, you say and we can see that GBA performed better, but the values show otherwise.

6) Paragraph "Combining STRING edges with homology" from "Results" Section: You make a reference of Fig2 after speaking about Smin, but Fig2 shows only Fmax values.

7) Paragraph "Effect of individual STRING data sources" from "Results" Section: You make a reference of Fig S5-8, they do not exist, Fig S2-4 do.

Reviewer #2: The authors have performed a very thorough evaluation of impact of PPI network sparseness on the performance of various GO term inference in PPI networks of 4 organisms. They have evaluated both the impact of the addition of edges from different sources to the network (both experimental as well as computationally inferred), and the usage of different strategies for inferring GO annotations from these networks (Guilt by association, sequence similarity, …). The analysis has been carefully performed, and represents a considerable amount of work to evaluate the impact on the prediction performance. I would have some questions and minor comments, as well as a suggestion for a more extensive evaluation which would in my opinion add an additional layer to this analysis

Suggestion for major improvement:

As the authors state, the available PPIs represent a tiny proportion of the full set of unknown interactions between proteins. However, this sampling is not an unbiased, random sampling, but is likely influenced by the fact that some proteins have been more studied than others. For example in human, oncogenes/proteins are much more likely to appear as hub proteins than other proteins, only because they have been the focus of more in-depth studies. Hence sparsity in one aspect, but biased sparsity is another important one.

Hence, I would suggest to add to the study an analysis to evaluate this effect. More precisely, I would suggest to take a relatively dense network like the yeast PPI, and through sub-sampling, obtain more and more sparse networks and evaluate the effect of this down-sampling on the prediction accuracy. This subsampling could be done either by (1) random, unbiased sub-sampling, or (2) by a procedure that would remove edges with a probability that is inversely proportional to the connectivity of the nodes. Hence, highly connected nodes would be more likely to keep their edges, while less connected proteins would be more likely to loose edges, simulating a situation in which the network contains more hubs. It would be interesting then to follow the decrease in prediction accuracy as more and more edges are removed by either of these 2 procedures.

Questions/minor points

- How is the GO hierarchy dealt with in this study? As the GO ontology contains a high number of terms, very often 2 proteins might be annotated to different terms, which are however very closely related in the hierarchy. Would they count as mismatches in this case? Did the authors use a simplified version of the GO terms (GO slim)? This should be more carefully explained!

- The authors should explain the definition of Fmax and Smin, as the readers might not be familiar with these evaluation metrics.

- Related to this, the use of Fmax and Smin has been questioned in a recent paper (Plyusnin et al., PLOS Comp. Biology 2019); could the authors comment on this? I understand that they used Fmax and Smin as these were the metrics used in the CAFA assessment, however I would like to have some comments on the performance of these metrics and the possible biases.

- The authors have evaluated the effect of using the node2vec procedure instead of the naïve GBA procedure, which should have the advantage of using a larger neighborhood compared to GBA. They state that node2vec is the preferred method compared to GBA (line 268). However, even if the trend shows an increase in performance in Fig 3, the improvement seems hardly significant. Could the authors quantify the improvement of node2vec compared to GBA in Figure 3?

- Typo in line 139.

6. PLOS authors have the option to publish the peer review history of their article (what does this mean?). If published, this will include your full peer review and any attached files.

Reviewer #1: No

Reviewer #2: **Yes: **Carl Herrmann

---

## [Author Response · Author response to Decision Letter 0]

17 Sep 2020

First, we would like to thank the editor and reviewers for their constructive comments. They helped us to improve the manuscript considerably. We provide a point-by-point answer to each of them. We present answers to reviewer’s comments (in blue) in black and report changes to the manuscript in italic.

Editor

Specifically, the computational experiments to address the extend to which Protein Protein Interaction Networks are useful in predicting Gene Ontology terms in different species are well thought and designed. Moreover, the work is well organized and presented in an intelligible manner. Both expert reviewers provide useful suggestions (see below for their detailed reports) which can strengthen the conclusions derived from this work and/or improve presentation.

I would also like to stress that you should make sure that all figures and tables are appropriately cited in the text (main text and/or supplement). For example, Tables S12 and S13 in Supplementary Text S1 are not cited either in the main manuscript or in Text S1.

Changes in manuscript: We made sure that all supplementary tables and figures, including those added after revision are appropriately cited in both the main document and S1 Text.

Reviewer #1: 

The paper aims to perform an analysis of how PPI networks can help in the functional annotation of proteins. The authors also want to study the impact of the quality of a PPI (how well known they are). 

To do so, they first compare naive algorithms to annotate function with guilt-by-association (GBA) and node2vec (n2v) algorithms using PPI. Then they compare GBA on different PPI: they combine the PPI with different STRING networks. Finally they compare those algorithms with a deep-learning method based on sequences. They do those comparisons on four organisms: yeast, with a well-known PPI, and A. thaliana, E. coli and tomato which do not have a well-known PPI. 

The paper is well written and nice to read. 

Major

1) The way the "EXP" PPI is made: you chose to remove the nodes that were connected but without functional annotations. You say it yourself in the Discussion: this might have negatively influenced the performance. But you don't explain your choice of getting rid of those nodes. You could either let the nodes in the "EXP" PPI, or make another PPI with those nodes to see the difference. 

You use algorithms that use neighborhood, then if you modify the neighborhood you can't expect them to perform well.

Thanks for this remark and allowing us to elaborate on this point. First, do realize that if nodes do not have a functional annotation, it is not possible to evaluate predictions made for these nodes as there is no ground truth. Hence, all predictions are deemed false positives and, moreover, certain measures, such as the recall and F1 scores, are undefined. Consequently, we should not include these unannotated proteins in the test set. Second, take into account that these unannotated proteins do have some functions but that these have not yet been discovered. Now, let’s consider what happens when including these unannotated proteins in the training set: For the GBA and kNN methods these proteins simply lower the posterior probability of their test neighbors having any function (by transferring a label vector of only zeroes). It is possible that these unannotated proteins will have some of the neighboring functions (but undiscovered up till now). Therefore, we believe that we should not dilute the prediction signal by including them in the training set. A similar argument holds for our linear classifier. For the node2vec method this is indeed different as node2vec is an unsupervised feature extraction step that only depends on the network and not the functional annotations. In the discussion of the previous version of the manuscript we referred to this aspect. Together with the reviewer we indeed suspected that including the unannotated proteins also as nodes in the network would possibly lead to better features in the first step of the node2vec procedure, as it leads to a better neighborhood estimation. However, at the 2nd supervised training phase, these nodes should not be used for the same reasons as mentioned before. 

We do agree with the reviewer, we could have substantiated more the influence on a possibly improved 1st phase on the node2vec procedure. Therefore, we performed a new experiment where we ran node2vec on the entire experimental (EXP) network (including unannotated proteins) and then used the extracted (unsupervised) features of the annotated proteins only in the 2nd phase using a 5-fold cross-validation step as before. The training, validation and test splits in each fold were kept identical and we also repeated the hyperparameter optimization steps of node2vec as big changes in topology might call for different hyperparameter values. We repeated this experiment for all four species and compared the performance with that of the original node2vec which trained the (unsupervised) features on a network of only annotated proteins. The results are shown in Table R1. We found that the network becomes (indeed) less disconnected when including the unannotated proteins as nodes in the network. Also, we could make predictions for more proteins, leading to an increased coverage. However, the increase was only statistically significant for E. coli and A. thaliana (paired t-test over the five cross-validation folds). More importantly, we did not observe any significant differences in either performance metric. The latter observation hints to that these new predictions are not as accurate as for the rest of network. Together, this highlights that a lack of ground truth annotation data hinders function prediction algorithms.

Table R1: Fmax, Smin and coverage of the node2vec method on the EXP PPI network in four species when excluding and including proteins without any functional annotations during the feature learning step.

 Fmax Smin Coverage

 Annotated Proteins Only All Proteins Annotated Proteins Only All Proteins Annotated Proteins Only All Proteins

S. cerevisiae 0.50 ± 0.012 0.50 ± 0.010 30.05 ± 0.54 30.13 ± 0.32 0.99 ± 0.002 0.99 ± 0.001

E. coli 0.28 ± 0.011 0.29 ± 0.008 20.37 ± 0.41 20.47 ± 0.31 0.77 ± 0.016 0.79 ± 0.017

A. thaliana 0.23 ± 0.008 0.23 ± 0.009 28.69 ± 0.49 28.94 ± 0.49 0.57 ± 0.010 0.58 ± 0.011

S. lycopersicum 0.08 ± 0.007 0.08 ± 0.007 19.37 ± 0.89 19.36 ± 0.89 0.03 ± 0.017 0.03 ± 0.017

Changes in manuscript: We have added a description of this experiment in the Methods section. In the Experimental set-up section (subsection “Optimization of node2vec classification”) we now mention that the optimization was done separately for the networks that include unannotated nodes. In the results (subsection “Only the yeast experimental PPI network has acceptable function prediction performance”) we have added a paragraph describing the results above and saying that including those extra nodes does not lead to performance improvement along with a reference to Table R1 (S4 in S1 Text). We have changed the discussion where we originally stated this hypothesis to: “It is worth noting that node2vec can make predictions for nodes that have no annotated neighbors, as opposed to GBA, which helps to increase the coverage. Nevertheless, including unannotated proteins with known interactions during the node2vec feature learning step did not lead to better function prediction performance. This hints that -apart from the lack of known interactions- the lack of GO annotations for training proteins also has a considerable negative effect on the accuracy of function prediction algorithms.”

Also you consider STRING networks as "predicted interactions", but all of them are not:

- "neighborhood" just states if the genes occur repeatedly in close neighborhood in genomes (mostly prokaryotic). So this one is not a prediction, and might be useful only for E.coli.

- "co-occurence" shows the occurence of two genes across species, here again mostly prokaryotic species, and no prediction. 

- "homology" is a score that is not used "as is" in STRING scoring schemes, it is likely a BLAST used in other channels. 

- "text-mining" can not predict anything because it just extract information from published articles.

You do prediction by using this data as input to one of the algorithm you use, but STRING networks are not predicted.

The reviewer is indeed right, these features are not predicted. We were addressing that these features are interpreted by STRING as that a pair of proteins interacts physically or is involved in the same task. But, given the purpose of our manuscript we do understand that this might be confusing.

Changes in manuscript: We changed the wording in subsection “predicted interactions” (from line 97 onwards) to clarify this: “Besides the experimental evidence, STRING contains protein associations from 12 data sources in total: "neighborhood", "neighborhood transferred", "co-occurrence", "database", "database transferred", "experiments transferred", "fusion", "homology", "co-expression", "co-expression transferred", "text mining" and "text mining transferred". We use these data as features predictive of two proteins interacting and/or being functionally associated to add edges to the experimental network. We refer to these edges as "predicted edges".” 

Minor

2) You have a typo in the abstract: "Here, we tested to what extened" -> "to what extent" 

Changes in manuscript: We corrected the spelling error.

3) You cite STRING v10, but on your github you say you used STRING v11.

Thanks for catching this error! We do indeed use version 11.

Changes in manuscript: We now refer to the appropriate publication.

4) You chose to select only the 50% highest non-zero scores, have you tried other percent? Why 50%? 

The threshold of 50% was chosen arbitrarily without trying other values. This threshold can be seen as a hyperparameter of the algorithm that can be tuned using a validation set. Alternatively, all non-zero edges can be kept and we could put a weight to each edge equal to the certainty of that edge as indicated by STRING. Then each node can transfer its functions to its neighbors with a weight equal to the weight of the interaction, resulting in a weighted GBA. As this is more universal than the thresholding method, we performed new experiments with this weighted GBA, i.e. we repeated the experiment of testing all possible combinations of STRING data sources and replicated those across the four species. In general, the weighted network tended to perform slightly better for most combinations. Figure R1 shows this for E. coli. 

Figure R1: Fmax (left), Smin (middle) and coverage (right) of the binary STRING networks when performing GBA on the 50% top edges (x-axis) vs the weighted GBA (y-axis) for E. coli. Each dot corresponds to one of the 511 combinations of the 9 STRING data sources added to the binary experimental network. The black dashed line shows the y=x line to ease comparison.

However, the effect sizes were rather small for the top-performing combinations (Table R2). In yeast, there was barely any difference and the performances of the best combinations of each approach varied at the fourth significant digit. In E. coli and A. thaliana, there is a consistent improvement in both Fmax and Smin, while the tomato results are mixed. Together, these results, although mixed, hint that STRING edges can indeed contain functional signal at lower certainty scores too. We have now included this additional experiment in the main text to highlight the effect of the threshold. 

Table R2: Fmax, Smin and coverage of the top-performing combination of the EXP+STRING network when using either a binary network with the 50% most probable edges or a weighted one with all edges. Statistically significant differences (paired t-test, FDR < 0.05) are shown in bold.

 Fmax Smin Coverage

 Binary Weighted Binary Weighted Binary Weighted

Yeast 0.49 ± 0.005 0.49 ± 0.004 27.43 ± 0.57 27.40 ± 0.61 0.99 ± 0.001 0.99 ± 0.001

E. coli 0.46 ± 0.008 0.48 ± 0.009 17.07 ± 0.22 16.61 ± 0.35 0.99 ± 0.004 0.99 ± 0.003

Arabidopsis 0.48 ± 0.004 0.49 ± 0.005 24.12 ± 0.50 23.54 ± 0.50 0.98 ± 0.003 0.99 ± 0.001

Tomato 0.61 ± 0.045 0.64 ± 0.043 9.08 ± 0.89 9.23 ± 1.00 0.86 ± 0.020 0.91 ± 0.018

Changes in manuscript: We have added a supplemental section in S1 Text that shows the results in the form of figures like Figure R1 for all four species and Table R2 (Figures S3-S6 and Table S6). In the main manuscript’s methods section (subsection “Predicted interactions”) we also mention this additional experiment and the weighted GBA classifier. In the Results (subsection “Adding predicted edges is more useful than using a complex classifier”), we added the following text: “Using a weighted STRING network with all available interactions instead of a binary one lead to small performance improvements, but mainly for the combinations that performed less well. The effect sizes were rather small for the top-performing combinations. This shows that STRING edges possibly contain useful functional signal even at confidence levels lower than those we considered here.”

5) In Table S2, you might have swap the "EXP, GBA" and "EXP, node2vec" tomato values, because on the text and on Figure 2, you say and we can see that GBA performed better, but the values show otherwise. 

Thanks for spotting this mistake! The performance of both GBA and node2vec in tomato is 0.08. This is essentially the lowest possible value, as it arises from assigning all terms to all proteins and signifies that both methods fail completely in this case. The absence of an orange bar in Figure 2d was caused by the fact that the orange and blue bars are at the same height, which was not made clear. The Smin performance of the two methods was also nearly identical in tomato (19.39±0.91 and 19.37±0.89, Table S3).

Changes in manuscript: We have corrected the value in Table S2 in S1 Text and adapted the caption of Figure 2 to explain that the bars are at the same height.

6) Paragraph "Combining STRING edges with homology" from "Results" Section: You make a reference of Fig2 after speaking about Smin, but Fig2 shows only Fmax values. 

Changes in manuscript: We changed the reference to Table S3 which contains the Smin values.

7) Paragraph "Effect of individual STRING data sources" from "Results" Section: You make a reference of Fig S5-8, they do not exist, Fig S2-4 do.

Changes in manuscript: We now refer to the correct figures S2-S4.

Reviewer #2 

The authors have performed a very thorough evaluation of impact of PPI network sparseness on the performance of various GO term inference in PPI networks of 4 organisms. They have evaluated both the impact of the addition of edges from different sources to the network (both experimental as well as computationally inferred), and the usage of different strategies for inferring GO annotations from these networks (Guilt by association, sequence similarity, …). The analysis has been carefully performed, and represents a considerable amount of work to evaluate the impact on the prediction performance. I would have some questions and minor comments, as well as a suggestion for a more extensive evaluation which would in my opinion add an additional layer to this analysis. 

Major

1) As the authors state, the available PPIs represent a tiny proportion of the full set of unknown interactions between proteins. However, this sampling is not an unbiased, random sampling, but is likely influenced by the fact that some proteins have been more studied than others. For example in human, oncogenes/proteins are much more likely to appear as hub proteins than other proteins, only because they have been the focus of more in-depth studies. Hence sparsity in one aspect, but biased sparsity is another important one. 

Hence, I would suggest to add to the study an analysis to evaluate this effect. More precisely, I would suggest to take a relatively dense network like the yeast PPI, and through sub-sampling, obtain more and more sparse networks and evaluate the effect of this down-sampling on the prediction accuracy. This subsampling could be done either by (1) random, unbiased sub-sampling, or (2) by a procedure that would remove edges with a probability that is inversely proportional to the connectivity of the nodes. Hence, highly connected nodes would be more likely to keep their edges, while less connected proteins would be more likely to loose edges, simulating a situation in which the network contains more hubs. It would be interesting then to follow the decrease in prediction accuracy as more and more edges are removed by either of these 2 procedures. 

Thanks for this interesting insight. We fully agree with the reviewer that besides the sparsity aspect, the PPI networks bear a research-based bias. The reviewer does suggest a very interesting additional experiment to study the influence of this bias, which we were happy to include in the new manuscript. As suggested, we started from the full experimental PPI network of yeast and step-wise removed 10%, 20%, 30%, …, 90% and 99% of its edges. The edges were removed at random, using the two strategies suggested by the reviewer. In the first strategy (“uniform”), the probability of removing an edge was uniform over all present edges. In the second strategy (“degree”), the probability of edge removal was inversely proportional to the smaller degree of the two nodes that the edge connects, so that nodes with the fewest edges are most likely to lose them. This procedure gave us 11 down-sampled networks per sampling strategy on which we ran the GBA classifier. This was repeated 5 times with different random seeds to obtain variability estimates.

We found that the Fmax is relatively robust to uniform edge removal up to 40-50%, but Smin deteriorates more quickly, meaning that predicting more specific terms suffers even under these simplified circumstances (Figure R2). The coverage also drops very slowly (at least initially), which implies that -as expected- most edges are removed from “dense” parts of the network so that the remaining edges can largely make up for this loss. In the “degree” sampling strategy, which is more realistic, we observed a much steeper drop for all three metrics. In this case, poorly-studied proteins lose their connections very quickly making it impossible to make predictions for them, as indicated by the steep decline in coverage. As a result, the average performance also reduces very fast.

Figure R2: Fmax (left), Smin (middle) and coverage (right) of the GBA method on the yeast experimental PPI network (y-axis) as a function of the fraction of missing edges (x-axis). Edges where removed at random, either 1) uniformly (yellow), or 2) inversely proportional to the node degree, so that least connected nodes are more likely to lose their edges (purple). Error bars show standard deviation of 5 rounds of random sampling. 

This experiment relates to Figure 2e of the main text, where we showed the performance of the GBA classifier (corrected for the naïve performance, which we refer to as Prediction Advantage (PA)) as a function of the fraction of known edges. We used the degree-based down-sampling of edges to test whether we could validate the linear relationship between the two quantities that we had observed previously. The new figure is shown below as Figure R3. Because each network is a superset of all networks with fewer edges due to the step-wise edge removal, these observations are largely dependent, making it difficult to fit a curve to them. However, from the Figure we can see that the linear trend was not validated by this simulation.

Figure R3: Prediction Advantage (PA) of Fmax as a function of the fraction of known interactions. Each species is shown as a blue dot and red line shows the least squares linear fit. PA values calculated by downsampling the original yeast network at different levels of missing edges are shown as green dots.

Changes in manuscript: We included this new experiment and figure R2 (as Figure S2) in S1 Text. We have added a paragraph in the Results (subsection “Only the yeast experimental PPI network has acceptable function prediction performance”) describing this experiment and the results: “To better characterize the effect of missing edges, we simulated the phenomenon in yeast by removing edges either uniformly at random or by an approach that makes nodes with the lowest degree more likely to lose their edges first (S1 Text). We found that the Fmax is relatively robust to uniform edge removal up to 40-50%, but Smin deteriorates more quickly, meaning that predicting more specific terms suffers even under this simplified missing edges scenario. The coverage also drops slowly (at least initially), which implies that most edges are removed from “dense” parts of the network so that the remaining edges can partly make up for this loss. In the degree-based sampling strategy, which is more realistic, we observed a much steeper drop for all three metrics. In this case, poorly-studied proteins lose their connections very quickly making it impossible to make predictions for them, as indicated by the steep decline in coverage. As a result, the average performance also reduces very fast. The PA values calculated from the degree-based down-sampling did not confirm the linear relationship between PA and fraction of known edges (green dots in Figure 2e).”

We also added the Prediction Advantage (PA) values calculated in this down-sampling experiment in Figure 2e that shows the PA with respect to the naïve approach as a function of known edges. In doing so, we also discovered a bug in the code that draws Figure 2e which lead to inverted values, which we corrected. This led to a slightly different slope (0.98 instead of 0.95) and p-value (0.016 instead of 0.049), but did not affect the conclusions drawn before from this figure.

Minor

2) How is the GO hierarchy dealt with in this study? As the GO ontology contains a high number of terms, very often 2 proteins might be annotated to different terms, which are however very closely related in the hierarchy. Would they count as mismatches in this case? Did the authors use a simplified version of the GO terms (GO slim)? This should be more carefully explained!

We did not use GO slim, but the whole GO version. GO annotations from the gaf file were propagated upwards to also include ancestral terms. This means that if a protein is annotated with a term and a classifier predicts a different but related term, it is likely that the two terms will have common ancestral terms. These will be counted as correct predictions. For example, if a protein is annotated “DNA metabolic process”, which has 13 ancestors excluding the root, but is predicted to be involved in “RNA metabolic process”, which has the exact same ancestors, we will have 13 True Positives, 1 False Positive and 1 False Negative. For the calculation of the F1 score, TPs, FPs and FNs are all treated equally, whereas for the Semantic Distance the weight of each term is determined by its information content. 

Note that the guarantee of making predictions consistent with the GO graph holds only for the GBA method and k-NN classifier in the case of node2vec. It does not necessarily hold for the ridge classifier. Although we did not correct the predictions of the ridge classifier to take the hierarchy into account, it outperformed k-NN for 2 out of 4 species, so we believe that this did not severely impact its performance.

Changes in manuscript: In subsection “GO annotations” (lines 136-138), we now make this explicit: “We used the entire GO graph (not restricted GO slim). Annotations were propagated towards the ontology root, so that when a protein is annotated with a term, it is also annotated with all its ancestors in the GO graph.”.

We also mention the possible inconsistencies produced by the ridge classifier (lines 183-185): “We did not post-process the predictions of the ridge method so it is possible that it makes predictions that are inconsistent with the GO hierarchy”.

3) The authors should explain the definition of Fmax and Smin, as the readers might not be familiar with these evaluation metrics. 

Changes in manuscript: We have added the definitions in S1 Text as well as a reference to those definitions in the methods section of the main document.

4) Related to this, the use of Fmax and Smin has been questioned in a recent paper (Plyusnin et al., PLOS Comp. Biology 2019); could the authors comment on this? I understand that they used Fmax and Smin as these were the metrics used in the CAFA assessment, however I would like to have some comments on the performance of these metrics and the possible biases.

The concerns raised by Plyusnin et al. about Fmax and Smin mainly have to do with these metrics being overly lenient to false positive predictions, which might be acceptable since many of the annotations are missing, meaning that most proteins are likely to be under-annotated. On the other hand, the same study showed that both metrics correlate very highly with the signal to noise ratio of the predictions. Based on that we argue that the limitations of the metrics probably will not alter the relative ranking of the tested methods, so our conclusions do not rely on the choice of evaluation measures.

Changes in manuscript: We have added a paragraph in the discussion pointing out this potential limitation of our work: “Although Fmax and Smin are the most widely-used evaluation metrics for function prediction, a recent study has raised concerns about them. The concerns, which were based on artificially generated predicted annotations, mainly have to do with these metrics being overly lenient to false positive predictions. This might not be a big problem, as due to missing annotations most proteins are likely to be under-annotated. The same study showed that both metrics correlate highly with the signal to noise ratio of the predictions. Based on that we argue that our conclusions do not rely on the choice of evaluation measures, but we believe that proper evaluation of function prediction algorithms is a pressing issue that requires further research.”

5) The authors have evaluated the effect of using the node2vec procedure instead of the naïve GBA procedure, which should have the advantage of using a larger neighborhood compared to GBA. They state that node2vec is the preferred method compared to GBA (line 268). However, even if the trend shows an increase in performance in Fig 3, the improvement seems hardly significant. Could the authors quantify the improvement of node2vec compared to GBA in Figure 3? 

Following the suggestion by the reviewer, we quantified the significance of the performance improvement of node2vec with respect to GBA. To do so, we binned proteins from each species based on their degree, as previously, and applied the Wilcoxon rank sum test when comparing the performance the two methods on the proteins of each bin. The resulting p-values were controlled for the False Discovery Rate (FDR) using the Benjamini-Hochberg method. We did that separately for each species. We used an FDR threshold of 0.05 and found that the differences between GBA and node2vec were statistically significant for all bins in A. thaliana and E. coli (with node2vec being the superior method). In yeast, node2vec is significantly better in 6 out of 9 bins, GBA in 1 out of 9 and in 2 out of 9 bins there are no statistically significant differences. Overall, these results indeed show that node2vec is the preferred method even when statistically compared.

Changes in manuscript: We have updated Figure 3 to denote significant bins with a star (*) next to the number of proteins in the bin. We have also changed section “Performance per protein” of the results to describe the statistical tests and the results. Finally, we have added a supplementary table (Table S4 in S1 Text) which lists the raw and corrected p-values with an appropriate reference from the main manuscript. 

6) Typo in line 139.

Changes in manuscript: Typo has been corrected.

---

## [Decision Letter · Decision Letter 1]

9 Nov 2020

A thorough analysis of the contribution of experimental, derived and sequence-based predicted protein-protein interactions for functional annotation of proteins

PONE-D-20-21508R1

Dear Dr. Makrodimitris,

We’re pleased to inform you that your manuscript has been judged scientifically suitable for publication and will be formally accepted for publication once it meets all outstanding technical requirements.

Kind regards,

Baldo Oliva

Academic Editor

PLOS ONE

Additional Editor Comments (optional):

Reviewers' comments:

Reviewer's Responses to Questions

**Comments to the Author**

1. If the authors have adequately addressed your comments raised in a previous round of review and you feel that this manuscript is now acceptable for publication, you may indicate that here to bypass the “Comments to the Author” section, enter your conflict of interest statement in the “Confidential to Editor” section, and submit your "Accept" recommendation.

Reviewer #1: All comments have been addressed

Reviewer #2: All comments have been addressed

2. Is the manuscript technically sound, and do the data support the conclusions?

Reviewer #1: Yes

Reviewer #2: (No Response)

3. Has the statistical analysis been performed appropriately and rigorously? 

Reviewer #1: Yes

Reviewer #2: (No Response)

4. Have the authors made all data underlying the findings in their manuscript fully available?

Reviewer #1: Yes

Reviewer #2: (No Response)

5. Is the manuscript presented in an intelligible fashion and written in standard English?

Reviewer #1: Yes

Reviewer #2: (No Response)

6. Review Comments to the Author

Reviewer #1: I would like to thank the authors who provided an enhanced version of the manuscript. The authors answered all my concerns, including my major one.

I am a bit surprised that the difference between all the nodes and only the annotated ones is that small. I suppose it depends a lot on the topology of the graph. I would have expect that on a sparse graph it could have improve the scoring because it would have create new links between annotated nodes (and not only "dilute" the signal).

Reviewer #2: (No Response)

7. PLOS authors have the option to publish the peer review history of their article (what does this mean?). If published, this will include your full peer review and any attached files.

Reviewer #1: No

Reviewer #2: **Yes: **Carl Herrmann

---

## [Editor Report · Acceptance letter]

13 Nov 2020

PONE-D-20-21508R1 

A thorough analysis of the contribution of experimental, derived and sequence-based predicted protein-protein interactions for functional annotation of proteins 

Dear Dr. Makrodimitris:

I'm pleased to inform you that your manuscript has been deemed suitable for publication in PLOS ONE. Congratulations! Your manuscript is now with our production department. 

Kind regards, 

on behalf of

Prof. Baldo Oliva 

Academic Editor

PLOS ONE